# Evaluation of High-Volume Injections Using a Modified Dorsal Quadratus Lumborum Block Approach in Canine Cadavers

**DOI:** 10.3390/ani12010018

**Published:** 2021-12-22

**Authors:** André Marchina-Gonçalves, Francisco Gil, Francisco G. Laredo, Marta Soler, Amalia Agut, Eliseo Belda

**Affiliations:** 1Departamento de Medicina y Cirugía Animal, Facultad de Veterinaria, Universidad de Murcia, 30100 Murcia, Spain; andre.marchinag@um.es (A.M.-G.); laredo@um.es (F.G.L.); mtasoler@um.es (M.S.); amalia@um.es (A.A.); 2Departamento de Anatomía y Embriología Veterinaria, Facultad de Veterinaria, Universidad de Murcia, 30100 Murcia, Spain; cano@um.es

**Keywords:** ultrasound, locoregional anesthesia, canine, abdominal analgesia

## Abstract

**Simple Summary:**

The quadratus lumborum (QL) block is an ultrasound-guided locoregional anesthesia technique. Its objective is to promote both visceral and somatic analgesia for abdominal procedures. Previous spread studies carried out in canine cadavers demonstrated its viability in this species but failed in consistently reach the spinal nerves responsible for the cranial abdominal wall innervation. Therefore, we hypothesize that a modified QL block technique, based on the administration of a higher volume of solution (0.6 mL kg^−1^) in a dorso-medial position compared to the interfascial injection point between the QL and psoas minor muscles, could enhance its cranial spread, and promote a consistent distribution spread through the ventral branches of the spinal nerves and sympathetic trunk. For this purpose, a solution of dye/contrast was ultrasound-guide injected into six canine cadavers. The results were assessed through computed tomography and dissection, showing that the proposed technique is viable, safe, and stained the median and caudal abdominal nerves and the sympathetic trunk up to T13 consistently. However, our modified technique of QL block did not increase the cranial distribution of dye/contrast to the thoracic spinal nerves, and may not provide adequate somatic analgesia of the cranial abdominal wall.

**Abstract:**

The quadratus lumborum (QL) block targets the fascial plane surrounding the QL muscle providing abdominal somatic and visceral analgesia. The extension of its analgesic effects is a subject of research, as it could not cover areas of the cranial abdomen in dogs. This study assesses in eight thawed canine cadavers, the distribution of high-volume injections (0.6 mL kg^−1^ of a mixture of methylene blue and iopromide) injected between the psoas minor muscle and the vertebral body of L1. Anatomical features of the area of interest were studied in two cadavers. In another six dogs, QL blocks were performed bilaterally under ultrasound-guidance. The distribution of contrast was evaluated by computed tomography (CT). Hypaxial abdominal muscles were dissected to visualize the dye spread (spinal nerves and sympathetic trunk) in 5 cadavers. The remaining cadaver was refrozen and cross-sectioned. CT studies showed a maximum distribution of contrast from T10 to L7. The methylene blue stained T13 (10%), L1 (100%), L2 (100%), L3 (100%), L4 (60%) and the sympathetic trunk T10 (10%), T11 (20%), T12 (30%), T13 (70%), L1 (80%), L2 (80%), L3 (60%) and L4 (30%). These findings may suggest that despite the high volume of injectate administered, this modified QL block could not produce somatic analgesia of the cranial abdomen, although it could provide visceral analgesia in dogs.

## 1. Introduction

A quadratus lumborum (QL) block is an ultrasound-guided regional anesthesia technique that targets the fascial plane of the QL muscle and could provide a sensorial blockade of multiple areas of the abdomen [1]. In recent years, some anatomical and spread studies have been published with the aim of evaluating its feasibility in dogs [2,3,4,5], cats [6] and goats [7]. Veterinary literature also includes a case report of a cat [8].

The QL muscle extends from the last three thoracic to the last lumbar vertebrae [9]. The ventral branches of the spinal nerves responsible for the innervation of the abdominal wall extend from T10 to L3 [10] (T9–L3 [11]). These nerves run between the QL and the psoas minor (PM) muscles through the thoracolumbar area [12]. The communicating branches that connect the spinal nerves with the sympathetic trunk, responsible for the visceral innervation of the abdomen, are located close to the QL and PM muscles [13]. Therefore, the spread of a local anaesthetic solution within the fascial plane around the QL aims to desensitize the abdomen innervation, thereby inducing both somatic and visceral analgesia [3,12,14].

Several techniques to perform this block have been studied in canine cadavers, involving different injection sites and approaches in relation to the QL muscle and its related structures. Garbin et al. injected a mixture of dye and iohexol in the interfascial plane formed between the PM and QL muscles [2]. In another report [3], the same authors studied the distribution of the dye solution after its injection in a different site, located lateral to the QL muscle and medial to the thoracolumbar fascia, through two different approaches (transversal and longitudinal). In these studies, volumes of 0.15 to 0.3 mL kg^−1^ were injected per hemiabdomen. In another study, Alamán et al. [4] investigated the distribution of 0.5 mL kg^−1^ of injectate (methylene blue) administered between the QL muscle and the vertebral body of L1. The results of these studies showed a consistent spread in the L1, L2 and L3 spinal nerves and in the sympathetic trunk, although the spreading of dye in the last thoracic nerves was limited. More recently, Viscasillas et al. [5] evaluated the staining of the ventral branches of spinal nerves through a dorso-lateral to ventro-medial approach for the QL block, administering the injectate between QL and PM.

The objective of this study is to analyze, in canine cadavers, the pattern of distribution and the staining of the ventral branches of the spinal nerves and sympathetic trunk of a mixture of methylene blue and iopromide (50%) solution injected in a dorso-medial location compared with the approach described by Garbin et al. [2], between the PM muscle and the vertebral body of L1, at a volume of 0.6 mL kg^−1^ on each hemiabdomen.

Our hypothesis was that the proposed modification of the QL block technique would facilitate the distribution of the dye/contrast mixture within the targeted ventral branches of the spinal nerves, closer to the communicating branches and sympathetic trunk. This fact, together with the administration of high volumes of injectate, would enhance its cranial spread, promoting, more reliably, a sensory blockade of the cranial aspects of the abdominal wall.

## 2. Materials and Methods

This study was approved by the Biosafety Committee in Experimentation (CBE 433/2021) and the Ethical Committee for Animal Experimentation (CEEA 740/2021) of the University of Murcia. A total of eight canine cadavers of different breeds were employed in the study. The dogs were humanely euthanized for reasons unrelated to the study and did not present any visible anatomical changes in the vertebrae or in the abdominal wall that could affect the technique. Animals were frozen immediately after death and thawed 72 h before the study.

The study was divided into two phases (Figure 1). In the first phase, two cadavers were used for anatomical dissection. To perform this, cadavers were placed in lateral recumbency and received an extensive trichotomy of the abdominal region. The skin of the abdomen was removed, and the abdominal musculature dissected to observe the different muscle layers: external oblique, internal oblique and transverse abdominal. The costoabdominal (T13), cranial iliohypogastric (L1), caudal iliohypogastric (L2), ilioinguinal (L3) and lateral femoral cutaneous (L4) nerves were dissected/identified inside the interfascial plane between the transverse abdominal muscle and the internal oblique muscle. The same procedures were repeated on the contralateral side.

Then, the animals were placed in dorsal recumbency and eviscerated after the incision of the linea alba. The path of the same nerves was then observed in the retroperitoneal region, to establish their location and its relationship with the QL, PM, psoas major muscles and with the sympathetic trunk, before piercing the aponeurosis of the transverse abdominal muscle. Two researchers (Francisco Gil and André Marchina-Gonçalves) performed all the dissections.

For the second phase of the study, six cadavers were used (one injection per each hemiabdomen). The animals were placed in lateral recumbency, and a wide trichotomy of the abdominal region was performed. Then, 0.6 mL kg^−1^ of a 50/50 solution of methylene blue (Panreac Quimica, AppliChem, Castellar del Vallès, Spain) at a concentration of 0.5% and radiopaque contrast medium (Iopromide 300 mg mL^−1^) (Utravist300, Bayer, Berlin, Germany) was injected on each abdominal side. Ultrasound-guided injections, using an ultrasound equipment with a linear array of 3–13 MHz (MyLab Gamma, Esaote, Florence, Italy), were performed. For this purpose, sonovisible needles (Ultraplex 10 mm, 30°, BBraun, Melsungen, Germany) were used. The same researcher (Eliseo Belda) performed all of the injections.

Computed tomography (High speed dual; General Electric, Health Care, Madrid, Spain) scans were carried out immediately after the injections. The cadavers were placed in sternal recumbency and transverse CT scans, at a thickness of 3 mm, were obtained from T8 to S1 by selecting bone and soft tissue algorithms. Reformatted images were obtained and evaluated by two radiologists (Marta Soler and Amalia Agut) in the dorsal and sagittal planes to assess the location and distribution of the contrast medium in all directions: medial-to-lateral, dorsal-to-ventral and cranial-to-caudal, and to establish a correlation with the posterior dissection findings.

After the CT scans, all the cadavers but one (5/6) were placed in dorsal recumbency, an incision was made in the linea alba, and a sternotomy was performed. The organs and cavities (abdominal and thoracic) were observed for the presence of dye. After removal of the organs, the dye distribution in large abdominal vessels, diaphragm and retroperitoneal space was analyzed. Ventral branches of the spinal nerves were individually dissected and evaluated macroscopically to assess the presence of staining and its location. The intercostals (T10, T11, T12), costoabdominal (T13), cranial iliohypogastric (L1), caudal iliohypogastric (L2), ilioinguinal (L3), genitofemoral (L3), lateral femoral cutaneous (L4) and femoral (L4–L6) nerves were evaluated. Finally, the abdominal great vessels and the PM muscle were removed to observe the presence and extension of dye along the sympathetic trunk. Three researchers (Francisco Gil, André Marchina-Gonçalves and Eliseo Belda) performed all the dissections and, later, an analysis of the dye spread. It was considered that the staining of the target nerves in a length of 1 cm, for all quadrants, was adequate to induce a clinical blockade. Weakly stained nerves were considered as not dyed, because of the possibility that their coloring was an artifact produced during anatomical dissection.

Finally, the remaining canine cadaver (1/6) was refrozen at a temperature of −80 °C after the CT scan. Three days later, this cadaver was cross sectioned into 1 cm thickness sections. These sections were used for a topographic study of the dye distribution.

### 2.1. Ultrasound-Guided Injection Technique

All cadavers were placed in lateral recumbency, and the probe was placed transversally to the hypaxial musculature, caudal and parallel to the last rib, at the level of L1 (Figure 2). The probe was tilted in dorsal direction, until the transverse process of L1 and its acoustic shadow was located. Another reference point searched before the injections were the aponeurosis of the transverse abdominal muscle and the hyperechoic line located between the QL and the PM muscles. In all hemiabdomens, the proposed injection point was located between the PM muscle and the vertebral body of L1.

Once the injection point was identified, the needle was slowly introduced “in plane” in a ventro-dorsal and latero-medial direction. The needle was advanced through the external and internal oblique muscles. At this point, the transverse abdominal aponeurosis was pierced, and the needle advanced through the QL and PM muscles until the tip of the needle reached the vertebral body of L1. Following this, the needle tip was placed between the ventro-lateral aspect of the vertebral body of L1 and the dorso-medial aspect of the PM muscle. A test injection with 0.5 mL of injectate was carried out. If the distribution of the injectate was considered adequate, the complete volume of dye/contrast solution was administered. The same technique of injection was repeated on the contralateral hemiabdomen.

### 2.2. Statistical Analysis

The statistical tests were performed using SPSS, version 24.0 (SPSS Inc., Chicago, IL, USA). Data are expressed as mean ± standard deviation (SD), median and range or number of animals as it was considered more relevant in any case. Normality was assessed using a Shapiro-Wilk test.

## 3. Results

### 3.1. Anatomical Study

Two dogs (Spanish Mastiff and Labrador Retriever) were used in the anatomical study. They weighed 42 and 28 kg, with a body condition score (BCS) of 4/9 and 5/9, respectively (Table 1).

The hypaxial muscles PM and psoas major were easily identified. The QL muscle was located dorso-laterally to these muscles, and extended from the last three thoracic vertebrae to the last lumbar vertebrae. From L1 to L4, the QL muscle was covered by the PM minor and then, caudally to L4 by the psoas major. Vertebral bodies delimited the QL dorsally, and the last two ribs and the transverse processes laterally.

At the thoracolumbar level, the ventral branches of the spinal nerves passed through the interfascial space located between the QL and PM muscles. Then, they were run laterally between the transversal fascia and the aponeurosis of the transverse abdominal muscle. Finally, they emerged and ran ventrally through the interfascial space located between the transverse abdominal muscle and the internal oblique muscle.

Below the PM and the greater vessels, the communicating branches, sympathetic trunk, and splanchnic nerves were identified. The communicating branches exited the spinal nerves shortly after emerging from the intervertebral foramen. These branches established communication between the spinal nerves and the sympathetic chain.

### 3.2. Injection Technique

The reference points for performing the technique were easily identified in most animals. The hyperechoic line and the acoustic shadow caused by the L1 transverse process were identified in 12/12 injections. The transverse abdominal muscle aponeurosis was identified in 10/12, while the interfascial plane between the QL and PM muscles was identified in 9/12. As the tip of the needle advanced, two “pops” were felt for 12/12 injections. The first “pop” corresponded with the transverse abdominal muscle aponeurosis, and the second with the interfascial plane between the QL and the PM muscles. In 4/12 injections, the tip of the needle could not be visualized. In these cases, the injectate was injected once the tip of the needle was in contact with the vertebral body of L1 (Figure 3 and Figure 4).

### 3.3. Tomographic Study

The contrast medium was detected in the target area for 100% of th hemiabdomens (12/12) (Figure 5). The spreading of contrast varied from T10 to L7. In 91.7% (11/12) of hemiabdomens, the contrast was observed as surrounding QL and PM muscles in all their quadrants from L1 to L3. In the remaining case, the contrast medium was found to only be located in the ventro-lateral aspects of both muscles and surrounding the caudal vena cava from T9 to L1. In 41.7% (5/12) of hemiabdomens a small amount of contrast was visualized into the retroperitoneal region and the posterior renal fascia. One injection resulted in the presence of contrast inside the transverse abdominal plane. In another case, small amounts of contrast were found in the epidural space, close to the intervertebral foramen of L2.

### 3.4. Spread Study

The cadavers used in this study were of different breeds: Yorkshire, Poodle, Belgian Malinois, Pointer and German Shepherd, with a mean weight of 19.1 ± 13.06 kg and BCS ranging from 2 to 5 out of 9 (Table 1). The dissection of the cadavers revealed an absence of solution in the abdominal and thoracic cavities, neither in abdominal nor thoracic organs. A wide distribution of dye was detected in the hypaxial muscles. Moreover, staining of the retroperitoneal region and the posterior renal fascia was observed in some animals. The QL and PM muscles were considerably stained in all their portions. Staining of the psoas mayor muscle was also more evidently observed for its cranial aspects.

The nerve-staining results are shown in Figure 6. A median of 4 (3–5) spinal nerves were stained on each hemiabdomen. None of the intercostal nerves evaluated (T10–T12) were stained. The costoabdominal nerve (T13) was stained in 10% (1/10) of the injections (1/10). The cranial iliohypogastric (L1), caudal iliohypogastric (L2) and ilioinguinal (L3) nerves were stained in 100% (10/10) of the injections (Figure 7). In one cadaver, the left ilioinguinal nerve was absent or could not be located. The lateral femoral cutaneous nerve (L4) and the genitofemoral nerve (L3) were stained in 60% (6/10) and 40% (4/10) of the hemiabdomens, respectively. The femoral nerves were not stained.

The dissection demonstrated the presence of dye in the sympathetic trunk and communicating branches after 8/10 injections in a median of 4 (0–7) vertebral segments. The staining of these structures occurred occasionally at a level of T10 10% (1/10), T11 20% (2/10), T12 30% (3/10) and L4 30% (3/10), and more consistently at the level of T13 70% (7/10), L1 80% (8/10), L2 80% (8/10) and L3 60% (6/10) (Figure 8). The injectate did not reach the sympathetic trunk in one of the cadavers (cadaver 4).

### 3.5. Cross-Section Study

A 6 kg and BCS 6/9 mongrel dog was used in this phase (Table 1). This analysis revealed a great distribution of dye through the PM muscles. The dye was also observed in the QL muscle and in the interfascial plane between both muscles. An accurate visualization of the individual structures of the sympathetic chain was not possible. Nevertheless, the intense coloration of the adipose tissue located dorsally to the great vessels and ventrally to the body of the vertebrae was noticeable, and was considered to be highly suggestive of a good distribution of the injections along the sympathetic trunk region (Figure 9).

## 4. Discussion

The results obtained in this study showed that the administration of higher volumes (0.6 mL kg^−1^) of solution between the PM muscle and the vertebral body of L1 could be considered as a valid and viable technique to perform QL blocks in dogs. The point of injection employed here was situated at the dorsal and medial to the interfascial site, between the QL and PM muscles, which was proposed as an injection site by Garbin et al. [2] and Viscasillas et al. [5], and was more medial than the site of injection employed by Alamán et al. [4]. This technique consistently stained the ventral branches of L1, L2 and L3 (100%) and the sympathetic trunk (80%). Therefore, it could be a suitable alternative for surgeries carried out in areas of the median and caudal abdomen. Regarding the staining of the spinal nerves, our findings agree with previous studies carried out in canine cadavers, which observed staining in 100% of the ventral branches of L1, L2 and L3 nerves in a high volume (0.3 mL kg^−1^) [2] and lateral QL-Transversal approach [3] groups.

Contrary to our hypothesis, a further cranial spread of injectate was not evidenced, despite the modified point of injection and the higher volumes of injectate administered in this study. Similar to our findings, Garbin et al. [2,3], Alamán et al. [4] and Viscasillas et al. [5] were also unable to stain consistently the ventral branches of the thoracic spinal nerves T10, T11 and T12 responsible for the somatic cranial abdominal innervation [10]. Spread studies performed in human cadavers also failed in reaching the spinal nerves responsible for the upper portion of the abdominal wall [15,16,17]. Similar findings were reported in studies carried out in volunteers [18,19]. In our study, we selected volumes of 0.6 mL kg^−1^ of injectate per hemiabdomen to evaluate whether higher volumes than those used in previous studies [2,3,4,5] could stain more cranial thoracic spinal nerves. This volume, in a concentration of 0.25%, represents a total dose of 3 mg kg^−1^ of bupivacaine or ropivacaine, lower than convulsive [20,21] and lethal intravenous doses [22,23].

The major splanchnic nerves arise from the thoracic sympathetic ganglion at the level of T13, and the minor splanchnic nerves from the sympathetic chain at the level of the first lumbar vertebrae [24]. These nerves are involved in the visceral innervation of the abdomen [13,24]. The administration of higher volumes of injectate produced a slightly more consistent and extensive spread of the dye/contrast solution through the sympathetic trunk in our study compared to previous reports [2,3]. Moreover, performing the injections at a more dorso-medial position may have also increased the area of sympathetic-trunk staining. Alamán et al. [4] also reported wider areas of staining of the sympathetic trunk after injecting high volumes of injectate using a dorsal approach. Further studies investigating the administration of different volumes of injectate between the QL and PM muscles would help to clarify the influence of the volume and site of injection on the visceral and somatic analgesia provided by the QL block in dogs. The extent of the sympathetic trunk staining observed in our study could be compatible with an effective blockade that is able to provide visceral desensitization of the abdomen.

The staining of the thoracic sympathetic trunk reached the T10 level. The intercostal nerves were not stained in any of the cases, which could limit the efficacy of this block to provide somatic analgesia of the cranial abdomen. Some studies have described the relevant role of the endothoracic fascia in the distribution of solutions through the paravertebral thoracic space [16,25,26]. The sympathetic trunk is located ventrally to this fascia, whereas the intercostal nerves are found dorsally. It could be argued that the endothoracic fascia could block the flow of injectate between these compartments, explaining our findings. However, Alamán et al. [4] described the distribution of the dye along the sympathetic chain and the spinal nerves (T11–T13 and T12–T13, respectively), suggesting that the dye solution could travel through the paravertebral compartments.

The genitofemoral nerve was stained in 4/10 of the injections. The areas of innervation of this nerve are the inguinal region and the inguinal mammary glands [27]. Therefore, the technique of QL block described here could have limitations during inguinal procedures or caudal mastectomies. Staining of the lateral femoral cutaneous nerves (L4) occurred in 6/10 hemiabdomens. This nerve is responsible for innervating the skin of the cranial and lateral aspects of the thigh [28], and it is irrelevant for abdominal procedures. Finally, the absence of staining of the femoral nerves implies that our modified QL block technique would not induce motor or sensitive deficits of the pelvic limbs.

A computed tomography analysis revealed a caudal spread of the injectate from the injection point (L1) to L5–L7. The contrast was visualized more ventro-laterally and further away from vertebrae and spinal nerves at the lumbar caudal level. This may justify why the femoral nerves were not dyed. These findings are in agreement with those observed by Viscasillas et al. [5]. They found a similar caudal distribution of the contrast medium, with no staining of the lumbar plexus. A contrast medium was also observed in five injections (5/12) posterior to the renal fascia and retroperitoneal space. These findings could indicate that the injectate was able to spread ventrally through the transverse fascia despite its more dorsal injection. In another case (1/12), contrast was found inside the transverse abdominal plane, probably because the needle pierced the transverse aponeurosis, thereby creating a new path of spread. The presence of contrast in one epidural space could be due to the spreading of contrast through the path of the caudal iliohypogastric nerve. However, inadvertent insertion of the needle tip into the epidural space trough the intervertebral foramen could not be ruled out. Alamán et al. [4] also reported an epidural spread in some of the cadavers. This potential risk might be considered when selecting a dorsal approach to the QL block. As this finding was observed in only one case (1/12), the opening of the vertebral canal during the anatomical dissection of the cadavers was not considered in our study. Finally, in another case, a more ventro-lateral distribution of the injectate, which surrounded the caudal vena cava, was observed. It is probable that the injection was performed at a latero-ventral point to the PM muscle in this case, and the needle perforated the transversal fascia allowing the solution to reach the caudal vena cava. These findings may justify the use of atraumatic sonovisible needles, so as to reduce the risk of nerve or vascular puncture and to improve the visualization of the needle during the block.

The analgesic efficacy of the QL block was studied in a recent meta-analysis carried out in adult human beings [29]. This work concluded that the technique led to a reduction in opioid consumption and a better postoperative analgesia after a cesarean section and renal surgeries. The most promising results were obtained for the C-section, involving the manipulation of the lower abdominal wall. The renal procedures entering this meta-analysis involved minimal manipulation of the upper abdominal wall. Therefore, the efficacy of this block to provide upper abdominal analgesia should be evaluated in new and better-designed clinical studies. Bjelland et al. [30] analyzed the efficacy of QL block in humans for post-surgical analgesia after total abdominoplasty. In this report, when compared to the control group, the QL did not provide a significant improvement in the consumption of opioids, postoperative pain, postoperative nausea and vomiting, and chronic pain. However, Zhu et al. [31] obtained good results in 31 out of 32 patients undergoing open liver surgery who received a continuous anterior QL blockade.

Based on our results, the modified QL block technique described here could produce adequate analgesia for procedures involving caudal and median aspects of the abdominal wall and visceral manipulation; but it might not be able to provide somatic analgesia for procedures in the cranial abdomen. Further clinical studies are required to assess its analgesic efficacy in a real clinical setting.

Our study has several limitations. The researcher responsible for the injections (EB), also performed the dissections. This fact could bias the assessment of the dissections. The number of cadavers available for this research was limited, which may have influenced the results. The injections were evaluated in small and medium sized animals (2 to 28 kg). It is understood that the visualization of important ultrasonographic structures to guide the block could be impaired in larger dogs. The injection point selected in our study could be more difficult to access or visualise in detail in larger dogs. It is also important to point out that the QL block is a challenging technique. Therefore, it should be performed by clinicians who are experienced in ultrasound-guided techniques. Finally, the pattern of distribution of the injectate solution could differ in cadavers to live animals. Besides, a histological study has not been performed so as to assess the integrity of the muscular tissue after the thawing process. The refreezing process could also have caused some bias in the cross-section study. In addition, differences in physical and chemical properties of the injected mixtures of methylene blue and iopromide, compared to those of the local anaesthetics, may alter the patterns of distribution of the injectate along the area of interest.

## 5. Conclusions

The modified QL block technique that was evaluated in this study showed a consistent spread of the injectate between the L1 and L3 spinal nerves and the sympathetic trunk. Thus, it may be useful to provide analgesia for surgical procedures involving the median and caudal aspects of the abdomen of dogs. Further studies are needed to assess the efficacy of this technique in real clinical scenarios. Despite the increase in the administered volume of solution, and the use of a dorso-medial injection point, the described technique was unable to provide a consistent and effective staining of the last thoracic nerves. This finding could suggest its relative inefficacy as a somatic analgesia for procedures involving cranial aspects of the abdominal wall.

## Figures and Tables

**Figure 1 animals-12-00018-f001:**
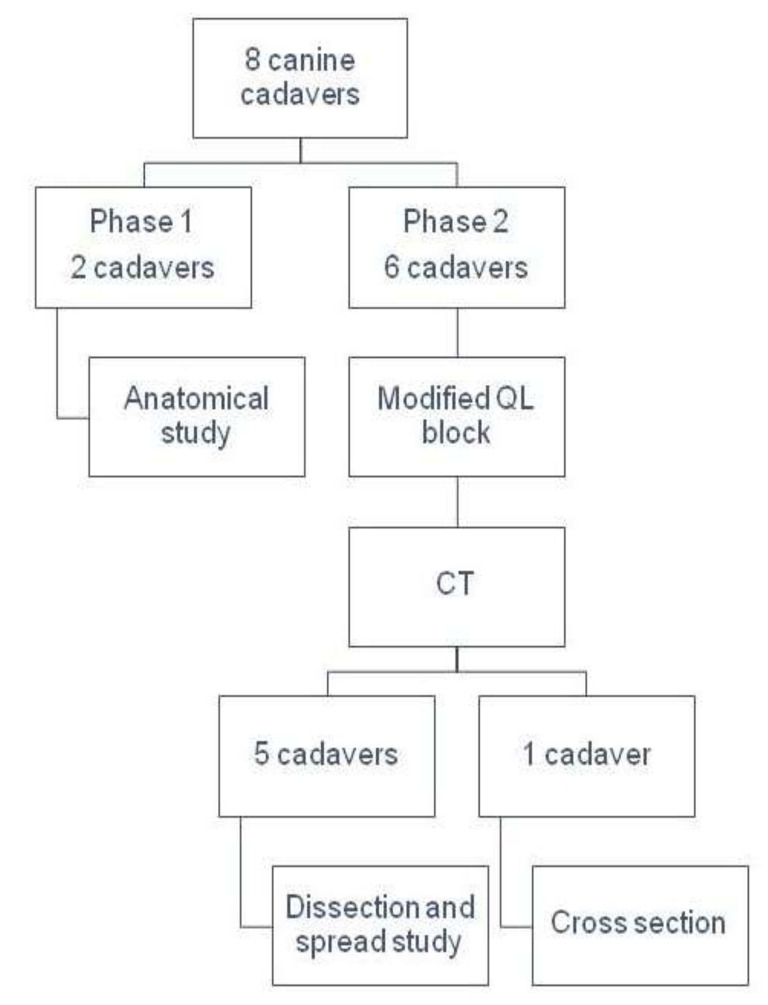
Schematic diagram of the experimental design. CT, computed tomography; QL, quadratus lumborum.

**Figure 2 animals-12-00018-f002:**
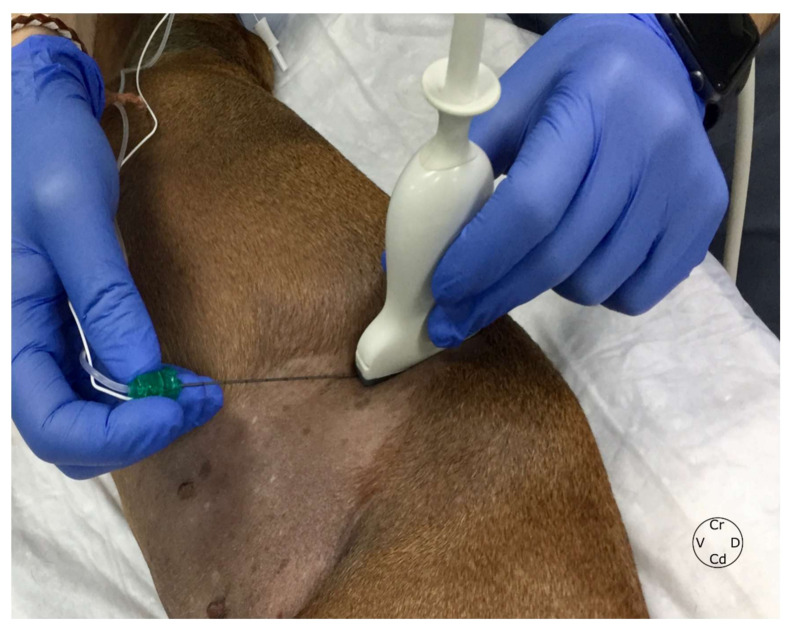
Ultrasound-guided approach to perform the QL block in dogs. Animals were positioned in lateral recumbency, and the transducer placed parallel to the last rib. The needle was inserted “in-plane” at the L1 level. QL, quadratus lumborum, L1; first lumbar vertebra, Cr, cranial; Cd, caudal; V, ventral; D, dorsal.

**Figure 3 animals-12-00018-f003:**
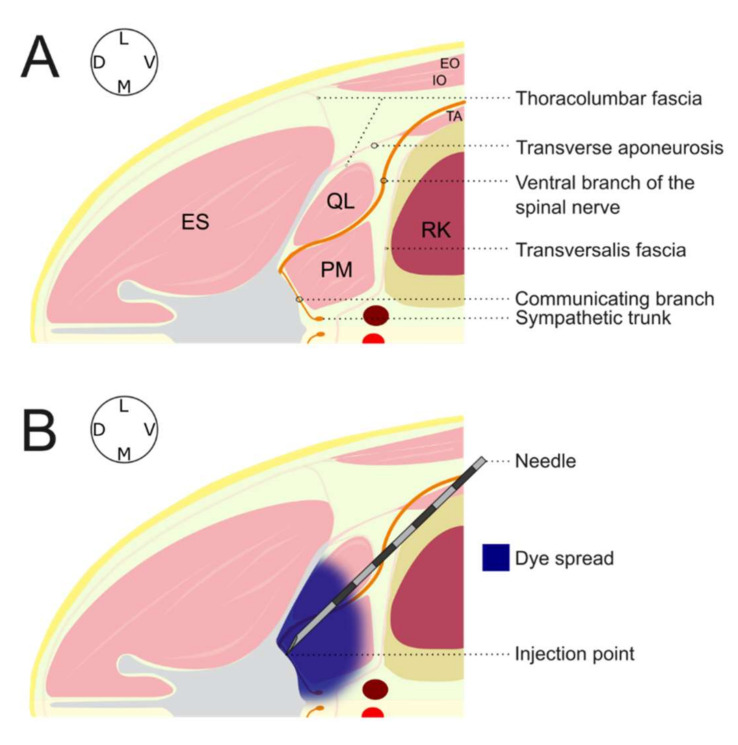
Schematic illustration of injection area at the L1 region in a dog placed in left lateral recumbency. (**A**) Main anatomical structures. (**B**) Final position of the needle and spreading of the injectate as observed in most hemiabdomens. L1, first lumbar vertebra; EO, external oblique muscle; IO, internal oblique muscle; TA, transverse abdominal muscle; ES, erector spinae muscles; QL, quadratus lumborum muscle; PM, psoas minor muscle; RK, right kidney; L, lateral; M, medial; D, dorsal; V, ventral.

**Figure 4 animals-12-00018-f004:**
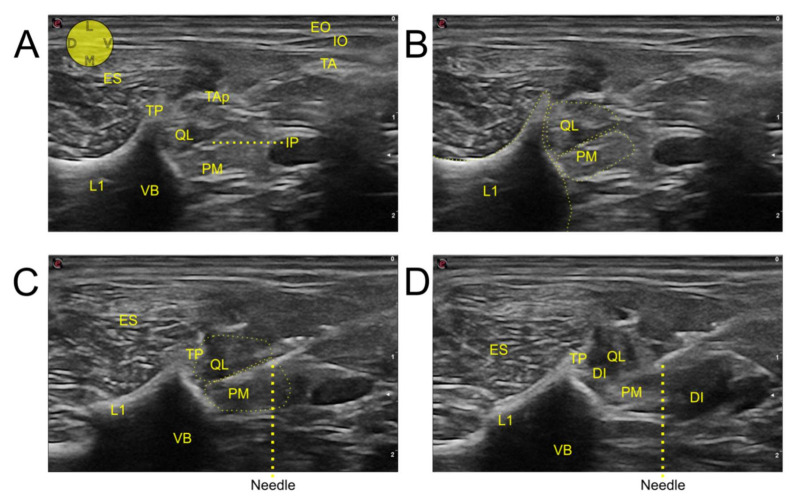
Ultrasound images of the modified approach to the QL block. (**A**) Sonoanatomy of the L1 region. (**B**) Schematic superimposition of the main anatomical structures. (**C**) Final position of the needle tip between the vertebral body and the psoas minor muscle. (**D**) Distribution of the injectate. L1, first lumbar vertebra; EO, external oblique muscle; IO, internal oblique muscle; TP, transverse process; TAp, transverse abdominal aponeurosis; TA, transverse abdominal muscle; ES, erector spinae muscles; QL, quadratus lumborum muscle; IP, interfascial plane; PM, psoas minor muscle; VB, vertebral body; DI, distribution of injectate; L, lateral; M, medial; D, dorsal; V, ventral.

**Figure 5 animals-12-00018-f005:**
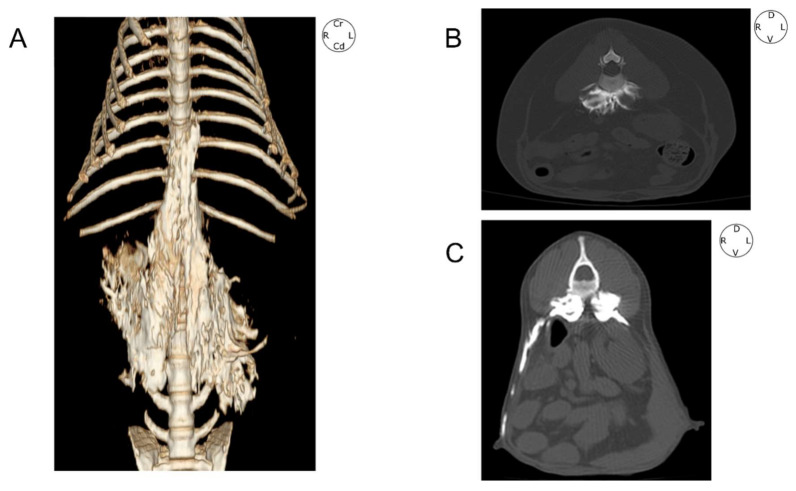
Computed tomographic images of the contrast spread after the administration of 0.6 mL kg^−1^ of a mixture of methylene blue and iopromide by a modified QL approach. (**A**) Volume-rendered three-dimensional projection image viewed from a ventral perspective of the thoracolumbar region. (**B**) Transverse image at L2 level with bone window setting. (**C**) Transverse image at L3 level showing contrasts presence in the left transverse abdominal plane. QL, quadratus lumborum block; L2; second lumbar vertebra; L3, third lumbar vertebra; Cr, cranial; Cd, caudal; L, left; R, right; D, dorsal; V, ventral.

**Figure 6 animals-12-00018-f006:**
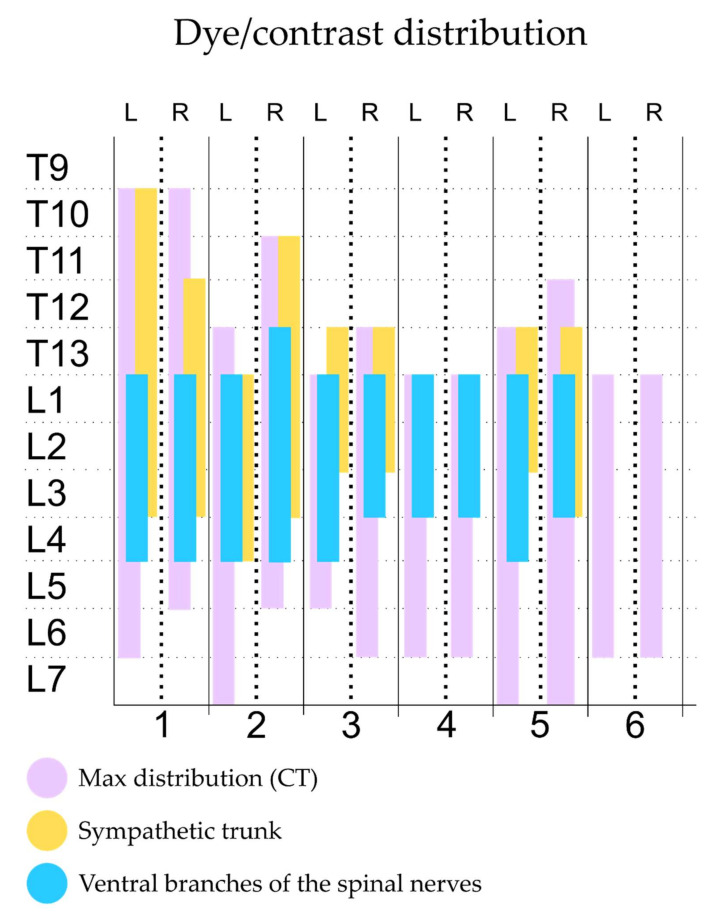
Staining of the ventral branches of the spinal nerves evaluated by computed tomographic and anatomical dissection after the administration of 0.6 mL kg^−1^ of a mixture of methylene blue and iopromide by a modified QL approach. QL, quadratus lumborum block; L, left hemiabdomen; R, right hemiadbomen.

**Figure 7 animals-12-00018-f007:**
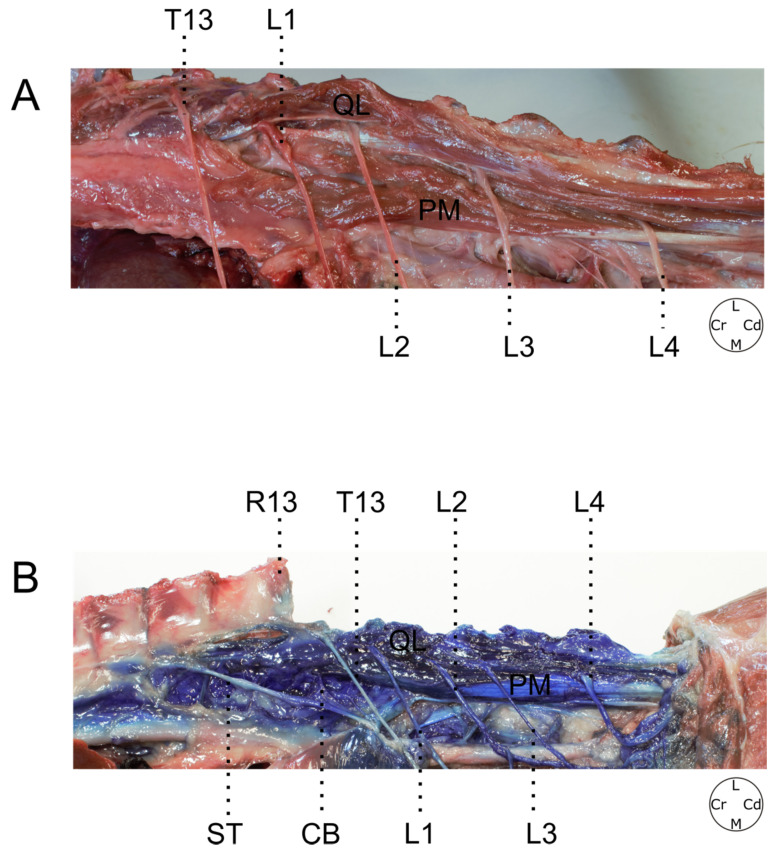
(**A**) Anatomical dissection of the thoracolumbar region. (**B**) Thoracolumbar region staining after methylene blue administration. T13; L1; L2; L3; L4, ventral branches of T13, L1, L2, L3 and L4 nerves respectively; QL, quadratus lumborum muscle; PM, psoas minor muscle; R13, 13th rib; ST, sympathetic trunk; CB, communicating branch; L, lateral; M, medial; Cr, cranial; Cd, caudal.

**Figure 8 animals-12-00018-f008:**
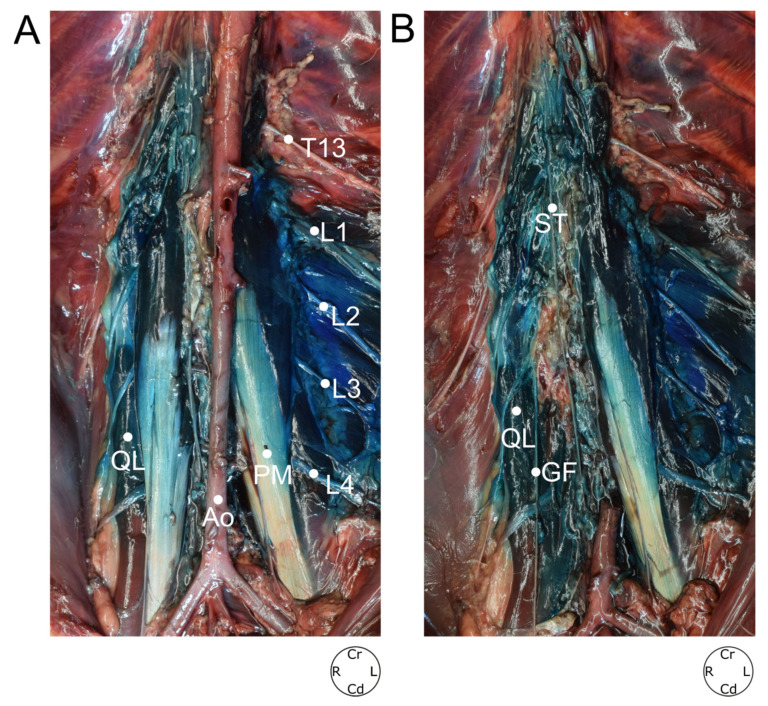
(**A**) Distribution of the dye through the quadratus lumborum and psoas minor muscles and spinal nerves. (**B**) Distribution of the dye through the lumbar sympathetic trunk after removing the right psoas minor muscle and the caudal aorta. T13; L1; L2; L3; L4, ventral branches of T13, L1, L2, L3 and L4 nerves respectively; GF, genitofemoral nerve; ST, sympathetic trunk; Ao, aorta; QL, quadratus lumborum muscle; PM, psoas minor muscle; Cr, cranial; Cd, caudal; R, right; L, left.

**Figure 9 animals-12-00018-f009:**
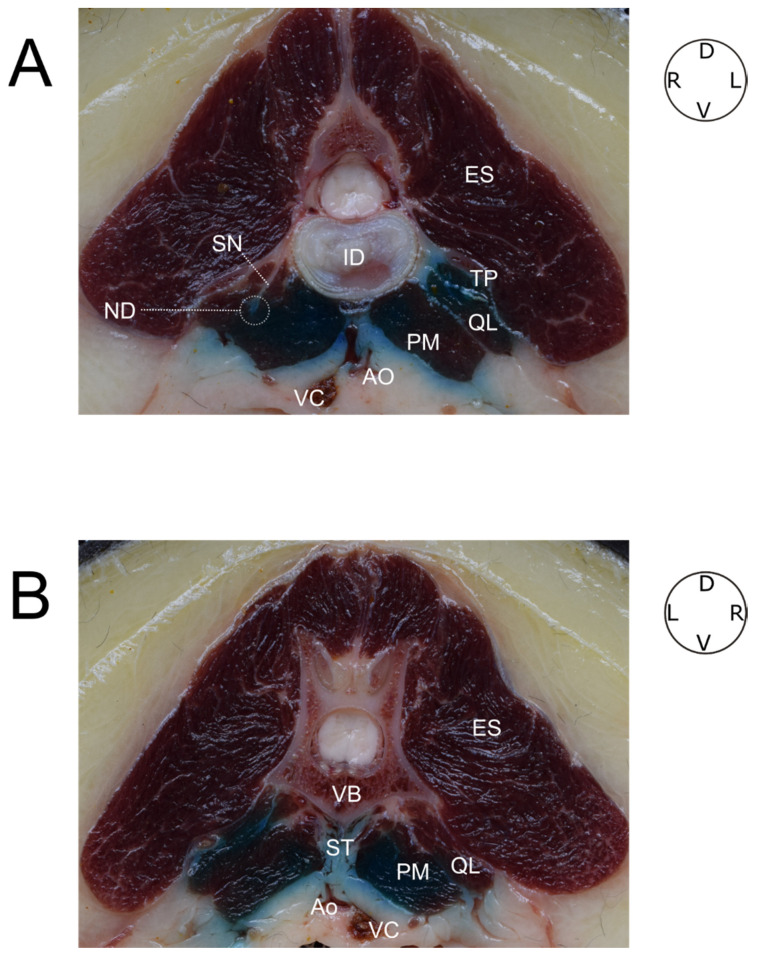
(**A**) Cranial view of a cross-section of the intervertebral disk and the caudal segment of L2. On the right side a spinal nerve can be observed leaving the intervertebral foramen and passing through the quadratus lumborum muscle. On the opposite side the presence of dye between the quadratus lumborum, the psoas minor muscles and the vertebral body can be observed. An intense coloration of the sympathetic trunk area between both psoas minor muscles is observed. (**B**) Dye distribution through the muscles and the sympathetic trunk area in a cranial vision of the center of L2. At this point, the spinal nerves cannot be seen. ES, erector spinae muscles; TP, transverse process; QL, quadratus lumborum muscle; PM, psoas minor muscle; ID, invetervebral disk; Ao, aorta; VC, vena cava; SN, spinal nerve; ND, nerve dyed; VB, vertebral body; ST, sympathetic trunk; D, dorsal; V, ventral; L, left side; R, right side.

**Table 1 animals-12-00018-t001:** Demographic distribution of the animals. BCS, body condition score.

Breed	Weight (kg)	BCS (1–9)
Spanish Mastiff	42	4
Labrador Retriever	28	5
Poodle	7.2	4
Belgian Malinois	22	2
Yorkshire	2.4	4
Pointer	22	4
German Shepherd	28	4
Mongrel	6.0	6

## Data Availability

Data supporting the reported results can be sent to anyone interested by contacting the corresponding author.

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
