# Peer review of "Evaluation of High-Volume Injections Using a Modified Dorsal Quadratus Lumborum Block Approach in Canine Cadavers"

_animals, 2021, doi:10.3390/ani12010018_

Round 1

Reviewer 1 Report

The method presented is an evolution of previous studies.

The multiple control methodologies all focused on the evaluation of
indirect drug diffusion are employed in a small number of patients and
have not been tested for repeatability.

The defreezing and subsequent freezing procedure for the cross section
may cause some bias in the evaluation of the diffusion of the dye.

There are no demonstrations regarding the tissue histological integrity
of the muscle which can influence the diffusion of the dye and the
final result may not be totally related to the injected volume.

These considerations could be correctly reported in the discussion  

Author Response

The method presented is an evolution of previous studies.

The multiple control methodologies all focused on the evaluation of 
indirect drug diffusion are employed in a small number of patients and 
have not been tested for repeatability.

The defreezing and subsequent freezing procedure for the cross section 
may cause some bias in the evaluation of the diffusion of the dye.

There are no demonstrations regarding the tissue histological integrity 
of the muscle which can influence the diffusion of the dye and the 
final result may not be totally related to the injected volume.

These considerations could be correctly reported in the discussion  

Dear reviewer, 

Thank you for your comments and suggestions. Your considerations have now been included in the limitations of the study:

“Our study has several limitations. The researcher responsible for the injections (EB), also performed the dissections. This fact could bias the assessment of the dissections. The number of cadavers available for this research was limited which may have influenced the results. The injections were evaluated in small and medium sized animals (2 to 28 kg). It is known that visualization of important ultrasonographic structures to guide the block could be impaired in larger dogs. The injection point selected in our study could be more difficult to access or to be visualized in detail in larger dogs. Finally, the pattern of distribution of the injectate solution could differ from cadavers to that which would occur in live animals. Besides, no histological study was performed to assess the integrity of the muscular tissue after the thawing process. The refreezing process could also have caused some bias in the cross-section study. In addition, differences in physical and chemical properties of the injected mixtures of methylene blue and iopromide compared to those of the local anaesthetics may alter the patterns of distribution of the injectate along the area of interest.”

Best regards, 

Authors

Reviewer 2 Report

The work of Marchina Gonçalves et al. performs a cadaveric study in which they present a modification of the anesthetic protocol of the quadratus lumbar with the aim of evaluating whether, by making a deeper injection and increasing the injected volume, somatic anesthesia is achieved in the caudal part of the thoracic region. 

This anesthetic blockade has received considerable attention in recent years and there are already several cadaveric studies that address various aspects of the technique, publications that are adequately collected and discussed in the present work.

Therefore, the study by Marchina Gonçalves et al., although not particularly novel, is well designed and adequately carried out. The anatomical study has been developed in detail, using various techniques that lead to consistent results.

Although the main hypothesis is not confirmed, since it is not possible to affect the thoracic somatic nerves, the results provide some detail with respect to previous works, since, in this case, a somewhat more cranial contrast distribution has been found than that seen in other articles, probably because of the volume used or because of the contrast administration site.

The discussion is adequate, also recognizing the limitations of the study. 

Some minor text editing:

On line 99, I would suggest changing “oblique internal” by “internal oblique”.

On Ref 25, the year is not indicated.

Author Response

Dear reviewer,

Thank you for your comments and suggestions. Your considerations have now been included in the revised text:

Point 1: On line 99, I would suggest changing “oblique internal” by “internal oblique”.

Response: Modified.

 “interfascial plane between the transverse abdominal muscle and the internal oblique muscle”

Point 2: 
On Ref 25, the year is not indicated.

Response: Completed.

“Portela DA, Campoy L, Otero PE, Martin-Flores M, Gleed RD. Ultrasound-guided thoracic paravertebral injection in dogs: a cadaveric study. Vet Anaesth Analg. 2017;44(3):636-645. doi:10.1016/j.vaa.2016.05.012”

Best regards,

Authors

Reviewer 3 Report

Dear authors,

Thank you for submitting the manuscript entitled “Evaluation of the pattern of distribution of high-volume injections of staining solution administered by a modified quadratus lumborum block technique in dog cadavers”. I found it to be an interesting study that can add valuable information to the literature. Nevertheless, I have some comments and suggestions that may enhance the final version of the manuscript.

Best regards!

General comments:

I honestly believe that it is the same approach as that described by Alaman et al. (2021) who recently described the dorsal approach quadratus lumbar.

The injection point in this modified approach is located between the PM and the vertebral body of L1. Considering the first sentence of the introduction ‘Quadratus lumborum (QL) block is an ultrasound-guided regional anaesthesia technique that targets the fascial plane of the QL muscle…’ You might think that it is not properly a QL block, right?

On the other hand, in the image of the cross-section study [Figure 9 (A and B)], it seems that the injection point could be that of the dorsal approach of the QL described by Alaman et al. (2021). The arrangement of the quadratus lumborum in this image is more in accordance with the description made by Evans & de Lahunta (2013) than the image provided in Figure 3 and the superimpositions of the muscular limits in Figure 4 (B and C).

Title:

I think the title is excessively long and should be summarized. Please use ‘canine cadavers’ instead of ‘dog cadavers’

Simple Summary:

Line 15: Terms like spread or distribution are more appropriate than "dispersion" when referring to the spread of the dye. Please amend them throughout the manuscript.

Line 16: Please use ‘species’ instead of ‘specie’.

Lines 17-18: Currently, there are three approaches described for the QL block in canine cadavers (Garbin et al. 2020a; Garbin et al. 2020b; Alaman et al. 2021) and the modified approach of one of them (Viscasillas et al. 2021). I think that if you talk about a modified approach, you should comment on which of them the modification has been made. In the same way regarding the injection point and the volume used. If it is a modification of an approach, it should be described in the M&M of Simple Summary and Abstract exactly how it is performed and the injection site.

Line 24: Substitute colorant for dye-contrast if that is the term you chose to use and keep it throughout the entire manuscript.

Abstract

Line 29: I think ‘thawed canine cadavers’ sounds better than ‘defrozen dog cadavers’.

Line 30: I would consider removing the abbreviation for methylene blue (MB) as it is not used too much.

Keywords

Keywords should be words not included in the title so that the search criteria for your work would be increased.

Ultrasound-guided locoregional anaesthesia is too long.

Introduction:

Line 45-47: Please add a reference for this statement.

Line 52: ‘in charge’ sounds weird. Please use better ‘responsible for’. Use the plural verb ‘extend’ instead of ‘extends’.

Line 56-59: The communicating branches and the sympathetic trunk are found in the dorsal and medial aspect of the QL and in the ventral and medial PM aspect, therefore the distribution of the solution in the plane between the QL and PM does not ensure visceral analgesia. Please rephrase

Line 56: ‘close to’ instead of ‘in close proximity’.

Line 61: All approaches described in the veterinary and human literature are described in relation to the QL muscle, not the PM muscle. Please remove 'and PM muscle'.

Line 62: When referring to cadavers, it is preferable to use the term inject than administer.

Line 66: ‘transversal’ instead of ‘transverse’.

Line 68: ‘of 0.3 and 0.5 ml kg-1’ or ‘a high volume solution of...’

Line 71: I think it would be more accurate to say that these nerves are stained, not that the dye is distributed between them. Viscasillas et al. (2021) did not assess the staining of the sympathetic trunk.

Line 75-76: As I said in the general comments the injection site is controversial to me.

Lines 74-77: You should better state the objectives of the study. Not only the distribution pattern of the solution is being evaluated (at the L1 level and assessed by CT), but also the staining of the spinal nerves and the sympathetic trunk. Please, reword this paragraph.

Line 80: spinal nerve roots or spinal nerve trunk?

Line 81: high volumes better than higher

Material & Methods:

Line 87: Why was the sample size eight cadavers and only six of them in the second phase? Please explain.

Line 92: I do not understand why the first phase of the study was carried out. What was your aim? To be familiar with the anatomical dissection or with the anatomy of the area? This whole part is widely described by Garbin et al. (2020a).

Line 103: ‘sought’ sounds weird to me. Please reword this.

Line 112: Any reference to justify this dye-contrast mixture?

Line 120: standard algorithms? Do you mean soft tissue algorithms?

Line 122: Please use the same name to refer to the solution and maintain it throughout the manuscript.

Line 135: EB is the researcher who performed the injections and also one of those who performed the dissections. It should be included in the study limitations.

Line 136: Why was the nerve considered stained with 0.5 cm of dye? In length or in all its quadrants? Please, could you explain it and add a reference?

Lines 144-149: I think the description of the approach should be more detailed specifying exactly the injection site.

Line 146: Attending to this description of the approach, how do you avoid that the acoustic shadow of the transverse process does not prevent visualizing the limits of the PM and QL?

Line 156-157: I think it should be described the entire needle pathway through the different muscles, not just QL and PM. At what point exactly in the vertebral body was positioned the needle tip? How did you know that the injection site was between VB and PM and not between VB and QL?

Results

Lines 174-187: This anatomical description has already been described and detailed by Garbin et al. (2020a) and it is not part of your objectives either. I would consider deleting it.

Line 190: ‘injections’ better than ‘cases’.

Line 193: In 4/12 of the injections the needle tip could not be visualized. So, how do you know the injection site was exactly between the PM and VB? The previous sentence is repetitive.

Line 194: In a third of the injections the needle tip was not visible. What is the explanation for it? Do you consider that this may be a problem using this approach?

Line 196-197: It doesn’t seem relevant information.

Line 215: "at the level of L1". The distribution pattern of the solution was only evaluated at the L1 level. This should be stated in the study objectives or provide more information on the pattern of distribution at different levels.

Line 219: It is very interesting. What explanation do you suggest for this?

Line 231: weight.

Line 262: Was this cadaver the same one in which distribution was observed in the transverse plane of the abdomen? In this case, could the injection be performed in a more lateral plane than the previous ones?

Line 306: Alaman et al. (2021) reported that 100% of the ventral branches of T13 and 30% of T12 were stained with their high volume solution (0.5 mg kg-1) and Garbin et al. (2020a) reported that 25% of T13 were stained. Please rephrase this sentence.

Lines 312-316: Summarize this paragraph in one sentence.

Line 317: arises.

Lines 328-330: The staining obtained in the study of the sympathetic trunk does not seem so consistent, T10 10%, T11 20%, T12 30% and T13 70%. Although it is a high percentage, one-third of individuals would not achieve the blockade of T13.

Lines 323-325: 1) Staining of the sympathetic trunk, not distribution. 2)...using a dorsal approach, it is not exactly a deeper point.

Using the dorsal approach described by Alaman et al. (2021), the sympathetic trunk staining is superior to yours with 5 (3-6) segments from T11 to L3. They achieve a 100% staining of T13 with a smaller volume than yours. How could these differences be explained?

Line 341: trajectory? I think is better to use ‘pathway’.

Lines 399-408: EB is the researcher who performed the injections and also one of those who performed the dissections. It should be included in the study limitations.

Line 412: Please, use ‘provide’ instead of ‘induce’.

References:

Reference number 2 could be eliminated since it corresponds to the presentation of an abstract in a conference of the authors who subsequently published reference 1 (Garbin et al. 2020a).

Figure Legends:

Figure 1. I think there are no groups in the study. There are only two phases. Please reword the figure legend.

Please, replace ‘Modified QL treatment’ with ‘Modified QL approach’. You have not performed any treatment. Please replace ‘dispersion study’ with ‘spread study’.

Define all abbreviations (CT, QL)

Figure 2. ‘Modified ultrasound-guided approach to perform a deeper QL block in dogs’. It should be specified on which approach the modification has been made. Please rephrase this sentence.

Define all abbreviations (CT, L1)

Table 1: Please, define all abbreviations using the same format in all table and image legends. Define QL.

Figure 3: Please amend in the figure: ventral branch of the spinal nerve, not spinal nerve; and dye spread not dye dispersion.

Figure 4: L1, first lumbar vertebra.

In figures a, b and d, the acoustic shadow of the transverse process does not allow the complete visualization of QL and PM. Would it be possible to get a better image?

In image c the limits of the muscles do not appear to be well established. The dorsal limit of the QL is the transverse process and the vertebral body (Evans & de Lahunta, 2013).

Figure 6: Substitute side for hemiabdomen. Please, delete the abbreviation MB. Define all abbreviations used (CT, QL).

Do you mean trunk of the spinal nerves, ventral branches or both?

Figure 7: Substitute the nerve names for L1, L2, ...

Please, delete the abbreviation MB. Define all abbreviations used.

Figure 9: Why was the cross-section done at the L2 level and not at L1?

Author Response

Dear reviewer,

Thank you for your comments and suggestions. We have tried to take into account all your considerations.

Please see the attachment for answers and comments.

Best regards,

Authors

Round 2

Reviewer 3 Report

Dear authors,

Thank you for accepting the suggested modifications. I think the manuscript has been improved and is ready to be considered for publication.

Best regards!

Author Response

Dear  reviewer,

Thank you for your comments, they certainly have improved our manuscript.  

We hope it could now be considered for publication.

Best regards,

Authors